# Inflammatory and neurodegenerative serum protein biomarkers increase sensitivity to detect clinical and radiographic disease activity in multiple sclerosis

Tanuja Chitnis [1], Ferhan Qureshi [2] ✉, Victor M. Gehman[2], Michael Becich [2], Riley Bove [3], Bruce A. C. Cree[3], Refujia Gomez[3], Stephen L. Hauser [3], Roland G. Henry [3], Amal Katrib[2], Hrishikesh Lokhande[1], Anu Paul[1], Stacy J. Caillier[3], Adam Santaniello [3], Neda Sattarnezhad[1], Shrishti Saxena[1], Howard Weiner [1], Hajime Yano[1] & Sergio E. Baranzini [3]

The multifaceted nature of multiple sclerosis requires quantitative biomarkers that can provide insights related to diverse physiological pathways. To this end, proteomic analysis of deeply-phenotyped serum samples, biological pathway modeling, and network analysis were performed to elucidate inflammatory and neurodegenerative processes, identifying sensitive biomarkers of multiple sclerosis disease activity. Here, we evaluated the concentrations of >1400 serum proteins in 630 samples from three multiple sclerosis cohorts for association with clinical and radiographic new disease activity. Twenty proteins were associated with increased clinical and radiographic multiple sclerosis disease activity for inclusion in a custom assay panel. Serum neurofilament light chain showed the strongest univariate correlation with gadolinium lesion activity, clinical relapse status, and annualized relapse rate. Multivariate modeling outperformed univariate for all endpoints. A comprehensive biomarker panel including the twenty proteins identified in this study could serve to characterize disease activity for a patient with multiple sclerosis.

Multiple sclerosis (MS) is a chronic inflammatory demyelinating disease of the central nervous system, with a variable presentation and heterogenous disease course[1,2]. While the exact pathophysiology of MS remains elusive, inflammatory and degenerative processes are believed to play a role[3–5]. Identifying disease-specific biomarker sets may assist in predicting diverse disease courses, classifying patients to high versus low risk for disease activity (DA) and progression (DP) and may also provide insights into mechanisms of new inflammatory DA[6,7]. Multivariate models reflecting multiple biological pathways involved in the complex

pathophysiology of MS will likely increase the predictive accuracy of these biomarkers[6].

Most studies have focused primarily on neurofilament light chain in blood serum (sNfL) as a biomarker in MS. Concentration of sNfL has been associated with neurodegeneration in MS and correlates with manifestations of DA, including the presence of gadolinium-enhancing (Gd+) lesions and clinical relapse[6–11]. For example, in a recent study of over 500 samples, a significant elevation in sNfL was observed after a clinical relapse only when associated with a Gd+ lesion[12]. In the 3 months after a Gd+ lesion, an average 35% elevation in sNfL

[1]Brigham and Women's Hospital, Boston, MA, USA. [2]Octave Bioscience, Inc, Menlo Park, CA, USA. [3]Department of Neurology. Weill Institute for Neurosciences. University of California San Francisco, San Francisco, CA, USA. ✉e-mail: fqureshi@octavebio.com

($p < 0.0001$) was reported when compared with samples from patients in remission[12]. Similarly, an average 32.3% elevation in sNfL was observed at the time of, or prior to a Gd+ lesion ($p = 0.002$) versus remission[12]. However, the observed increase in sNfL levels during a relapse has limited sensitivity and specificity and is of the same magnitude as the group-level coefficient of variation seen across populations of healthy individuals, thus rendering this metric insufficient for clinical decision-making[12,13].

The MS relapse disease process includes both an inciting inflammatory peripheral and central immune activation, with subsequent central nervous system damage in the form of myelin and neuronal degradation[1], as evidenced by immunological studies, and response to specific immune cell–targeting disease-modifying therapies. Moreover, MS genetic susceptibility studies demonstrate involvement of T and B cell–associated genes[14]. Thus, including additional inflammatory and neurodegenerative protein biomarkers, can offer deeper insights and reveal stronger correlations to clinical and radiographic DA than sNfL alone.

In the search for more specific and comprehensive sets of markers, several cytokines, chemokines, and other immune-related molecules have been associated with DA in patients with MS. For example, baseline levels of cerebrospinal fluid (CSF) proteins CXCL13, CXCL12, IFNγ, TNF, sCD163, LIGHT, and APRIL have been associated with evident DA compared with no evidence of DA (NEDA) in patients with MS[15]. CSF levels of glial biomarkers, including glial fibrillary acidic protein (GFAP) and Chitinase 3-like-1 (CHI3L1 or YKL-40), have been associated not only with DA but also with disability[16,17]. Masvekar et al. found that the additive model of IL12p40 and CHI3L1 correlated with new MS lesion activity[18]. MS disease severity has been associated with alterations in proteins reflecting astrocyte (MMP7, SERPINA3, GZMA, and CLIC1) and microglial activation (DSG2 and TNFRSF25)[19].

To provide insights and identify sensitive biomarkers of new MS DA, both radiographic (new Gd+ lesions) and clinical (relapses), we evaluated over 1400 serum proteins in over 600 samples from three MS cohorts. Biological pathway modeling and network analysis were performed to ensure comprehensive representation of MS neurophysiology and to gain insights into the inflammatory, immune, and neurodegenerative process in MS.

## Results

Results from this cross-sectional study are divided into the following three sections: protein (i.e., feature) selection analysis, univariate analysis of each endpoint outlined in the Methods, and multivariate modeling of each endpoint.

### Twenty proteins associated with disease activity were selected for the custom assay panel (CAP)

The final list of CAP proteins was selected by examining the univariate and multivariate associations of 1411 proteins with our three endpoints, constrained by a number of analytic and operational considerations. A detailed discussion of the process for arriving at this final list can be found in the Protein feature-selection section (and Table 1). GFAP was not one of the original proteins described in the Methods section. It was added to our panel after much of the development work described in this report was completed because of its strong association with several DA- and DP-related endpoints. Therefore, it was not part of the analysis. The remaining 20 proteins in Table 1 were carried forward.

### Univariate analysis identified several individual proteins significantly associated with Gd lesion regression, Gd lesion classification, and annualized relapse rate (ARR)

As a precursor to a formal univariate analysis of each endpoint, we grouped samples by label and represented the results as box plots to look for qualitative trends in concentration for each class at a population level. Gd lesion samples were separated into groups for zero, one, two, and three or more Gd+ lesions (Fig. 1). Purely binary endpoints include clinical relapse status (CRS, quiescence vs. exacerbation) and low versus high annualized relapse rate (ARR, ≤ 0.2/year for low and ≥ 1.0/year for high) were split according to positive and negative labels (Fig. 2).

To quantitatively investigate the univariate significance, Spearman's correlation and Student's $t$ test were used. The former checked for correlation between protein concentration and lesion count. The latter checked for differences in protein concentration means in the binary endpoints, univariate separation of samples associated with no Gd lesions from those associated with ≥1, as well as CRS and ARR status. Spearman's correlation and Student's $t$ test were computed for each protein in Fig. 3 to assess the relationship between protein concentration and all endpoints.

Examination of the bar charts in Fig. 3 allowed us to check the directionality and significance of all univariate statistical tests. Of particular interest were proteins showing consistency between the direction of the correlation/separation for all endpoints. Agreement between correlation and separation was particularly important for Gd lesion count and presence, since these two endpoints are not independent of each other. Proteins showing a test statistic with opposite signs between the two Gd tests did not pass the significance threshold. Furthermore, only CDCP1 shows a difference between the Gd endpoints and the other two, with negative correlation/separation for Gd lesions, but positive mean shift with clinically active disease state (CDCP1 shows no significant relationship to our ARR endpoint).

The proteins passing the significance threshold in each of the univariate analyses were (with test p-values): Gd lesion regression: (NfL [$3.3 \times 10^{-34}$], MOG [$1.6 \times 10^{-7}$], CDCP1 [$5.0 \times 10^{-3}$], CXCL9 [$8.4 \times 10^{-3}$], IL-12B [$2.4 \times 10^{-2}$], TNFSF13B [$2.7 \times 10^{-2}$], OPG [$2.8 \times 10^{-2}$], CCL20 [$4.2 \times 10^{-2}$]); Gd lesion classification: (NfL [$1.0 \times 10^{-19}$], MOG [$1.9 \times 10^{-6}$], APLP1 [$2.3 \times 10^{-3}$], VCAN [$9.9 \times 10^{-3}$], CDCP1 [$1.8 \times 10^{-2}$], CXCL9 [$4.7 \times 10^{-2}$]); CRS: (NfL [$5.0 \times 10^{-5}$], GH [$4.4 \times 10^{-3}$], SERPINA9

## Table 1 | Custom assay panel (CAP) proteins picked in the protein feature-selection process

| Analyte | Protein name/alias | UniProt ID |
|---------|--------------------|-----------|
| APLP1 | Amyloid β Precursor-Like Protein 1 | P51693 |
| CCL20 | C-C Motif Chemokine Ligand 20 (MIP-3) | P78556 |
| CD6 | Cluster of Differentiation 6 | P30203 |
| CDCP1 | CUB Domain Containing Protein 1 | Q9H5V8 |
| CNTN2 | Contactin 2 | Q02246 |
| COL4A1 | Collagen-α1(IV) Chain | Q07325 |
| CXCL9 | Chemokine (C-X-C Motif) Ligand 9 (MIG: Monokine Induced by γ Interferon) | O43927 |
| CXCL13 | C-X-C Motif Chemokine Ligand 13, BLC | P02462 |
| FLRT2 | Fibronectin Leucine-rich Repeat Transmembrane | O43155 |
| GFAP | Glial Fibrillary Acidic Protein | P14136 |
| GH | Growth Hormone, Somatotropin | P01241 |
| IL-12B | Interleukin-12 Subunit Beta | P29460 |
| MOG | Myelin Oligodendrocyte Glycoprotein | Q16653 |
| NEFL | Neurofilament Light Polypeptide Chain, NfL | P07196 |
| OPG | Osteoprotegerin, TNFRSF11B | O00300 |
| OPN | Osteopontin | P10451 |
| PRTG | Protogenin | Q2VWP7 |
| SERPINA9 | Serpin Family A Member 9 | Q86WD7 |
| TNFRSF10A | Tumor Necrosis Factor Receptor Superfamily Member 10A (TRAIL-R1), DR5, Death Receptor 5 | O00220 |
| TNFSF13B | Tumor necrosis factor superfamily member 13B, B Cell Activating Factor, BAFF | Q9Y275 |
| VCAN | Versican Core Protein, Versican Proteoglycan | P13611 |

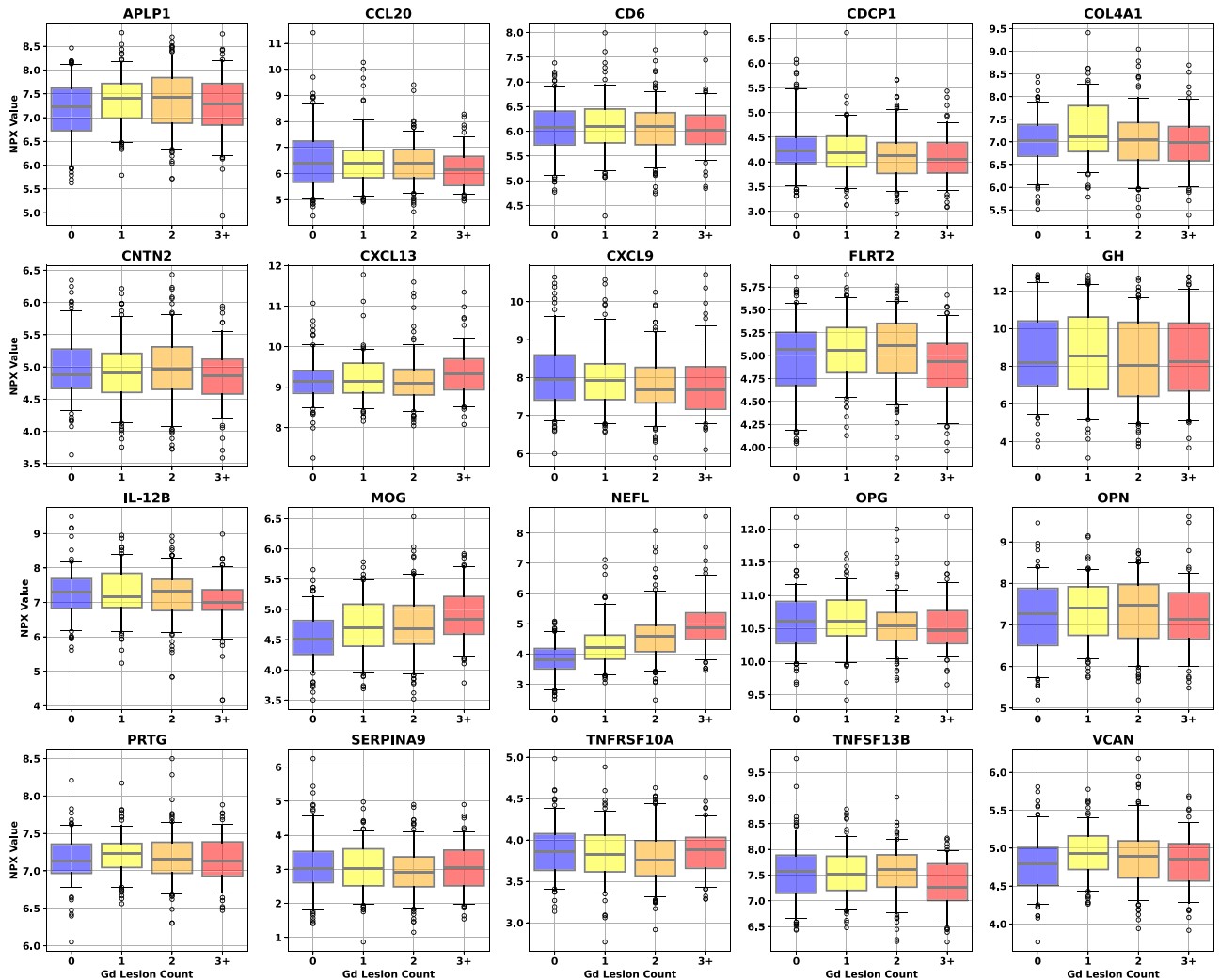

**Fig. 1 | Univariate dependence of the 20 CAP protein concentrations on Gd-enhanced lesion count.** Different color boxes correspond to the lesion count in that population of samples (blue for zero lesions, yellow for one, orange for two, and red for three or more). Sample counts for each lesion bin are 138 for 0 lesions, 126 for 1, 148 for 2, and 89 for 3 or more. The black line through each box shows the median (50th percentile) of the population. The height of each box shows the interquartile range (25th–75th percentile). The whiskers show the central 90% of the distribution (5th–95th percentile). The 5% of outliers furthest from the median are drawn as open black circles. Source data are provided as a Source Data file. CAP custom assay panel, Gd gadolinium, NPX normalized protein expression.

$[5.3 \times 10^{-3}]$, FLRT2 $[6.6 \times 10^{-3}]$, CDCP1 $[1.6 \times 10^{-2}]$, PRTG $[4.2 \times 10^{-2}]$); ARR: NfL $(7.9 \times 10^{-4})$.

## Multivariate models significantly outperform univariate models with NfL emerging as the strongest biomarker

Forward selection curves, regression, and classification Gd lesion analysis, as well as the classification of CRS and AAR are shown in Fig. 4.

Multivariate model performance for all four analyses, along with permutation-based feature importance were plotted in Fig. 5. Results for the Gd lesion regression analysis were presented as a two-dimensional histogram depicting the distribution of actual versus predicted lesion count for the regression analysis, and as receiver operating characteristic (ROC) curves for the classification analysis. The central line in each ROC band represented the mean of the ROC across all bootstraps; band width was the uncertainty in that mean. The regression analysis panel of Fig. 5 also included the best fit line through the results drawn in solid black with the root mean square error as a shaded gray region. Perfect agreement (actual = predicted) was drawn for reference in dashed gray. For the classification analysis panels, we included the following two feature sets in addition to the greedy forward selection (GFS) proteins for each endpoint: NfL only, and all features except NfL. Feature importance was estimated by randomly permuting the concentration values of each protein individually across samples and checking the performance decrease of the model for each bootstrap built on the GFS features.

For the Gd lesion detection endpoint, we quantified model performance in three different ways. General DA (GDA) was used for separation of samples with no Gd lesions from those with any positive count of lesions, Subtle DA (SDA) was used for separation of samples with no Gd lesions from those with only one lesion, and Extreme DA (EDA) was used for separation of samples with no Gd lesions from those with three or more lesions.

While we only trained lesion detection models on GDA, we included model performance metrics for the SDA and EDA endpoints as well in our discussion of Gd lesion detection. Additionally, we checked Gd lesion regression performance on models constructed from both NfL only and all proteins except NfL for comparison to the classification endpoint. The GFS feature models performed as follows: the square of Pearson's correlation coefficient $(R^2) = 0.280 \pm 0.027$ for Gd lesion count regression, area under the receiver operator characteristic (AUROC) $= 0.813 \pm 0.015$ for GDA classification, AUROC $= 0.845 \pm 0.026$ for CRS classification, and AUROC $= 0.803 \pm 0.039$ for ARR status classification. We tabulated the performance of all multivariate analyses in Table 2.

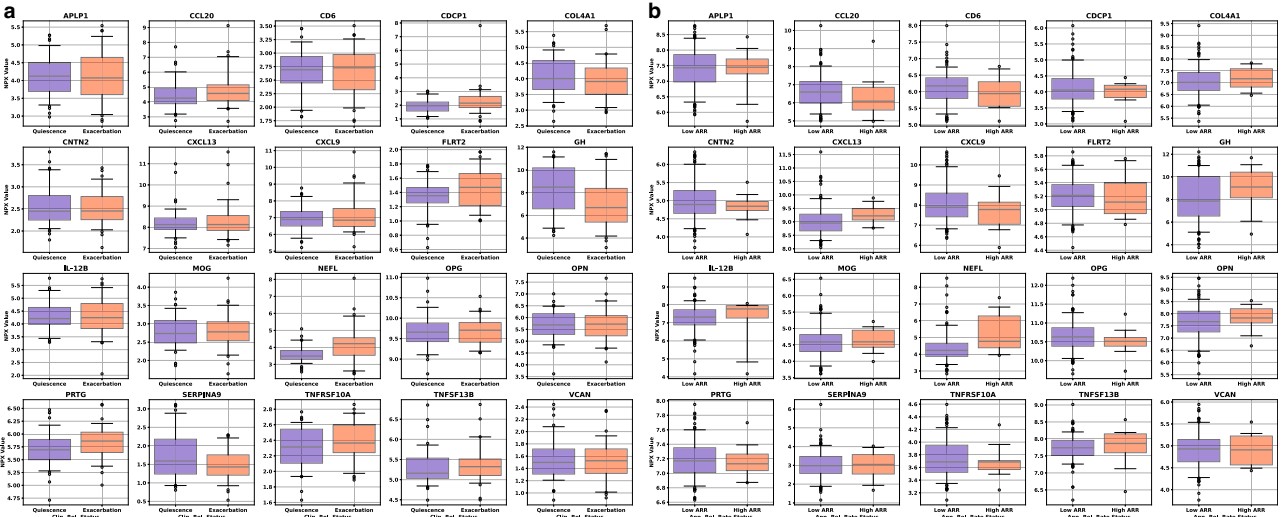

**Fig. 2 | Univariate box plots for the CAP proteins separation of samples (a) and box plots of the univariate separation of CAP proteins for low and high ARR (b). a** Univariate box plots for the CAP proteins separation of samples taken during quiescence (remission, blue boxes, 64 samples) or exacerbation (relapse, red boxes, 60 samples). **b** Box plots of the univariate separation of CAP proteins for low ( ≤ 0.2/year, blue boxes, 148 samples) and high ( ≥ 1.0/year, red boxes, 13 samples) ARR. The black line through each box shows the median (50th percentile) of the population. The height of each box shows the interquartile range (25th–75th percentile). The whiskers show the central 90% of the distribution (5th–95th percentile). The 5% of outliers furthest from the median are drawn as open black circles. Source data are provided as a Source Data file. ARR annualized relapse rate, CAP custom assay panel.

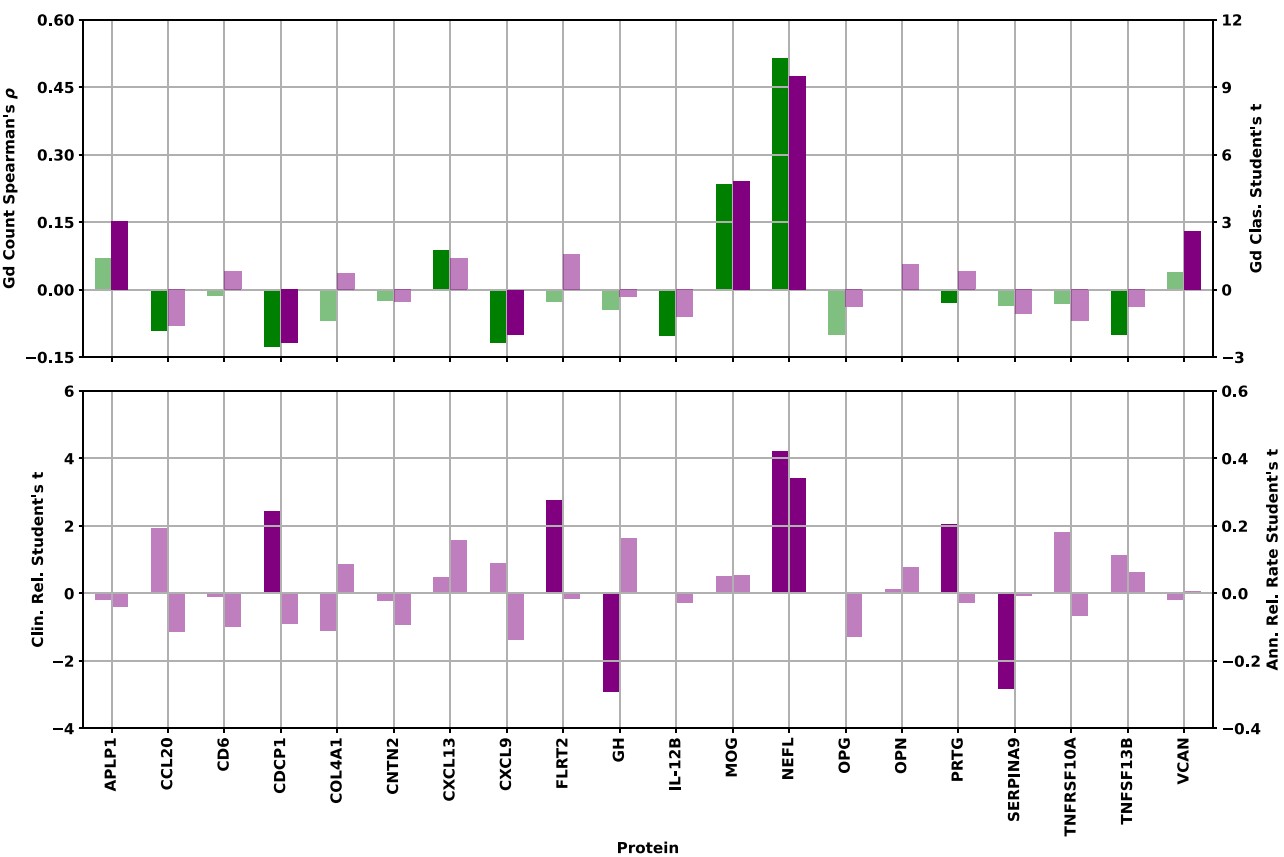

**Fig. 3 | Univariate statistical tests for all endpoints.** Top: Spearman's ρ correlation between NPX concentration and Gd lesion count (green bars, left axis) and Student's t statistic (two-sided) for separation of samples associated with zero lesions from those with one or more by NPX concentration (purple bars, right axis) for each protein. Bottom: Student's t statistic (two-sided) for separation of samples associated with clinically inactive from those with clinically active disease state (left axis) and those associated with low from high ARR (right axis) by NPX concentration. Bars corresponding to statistical tests showing a *p*-value > 0.05 have been drawn in a lighter shade of the same color to denote their lack of statistical significance. Source data are provided as a Source Data file. ARR annualized relapse rate, Gd gadolinium, NPX normalized protein expression.

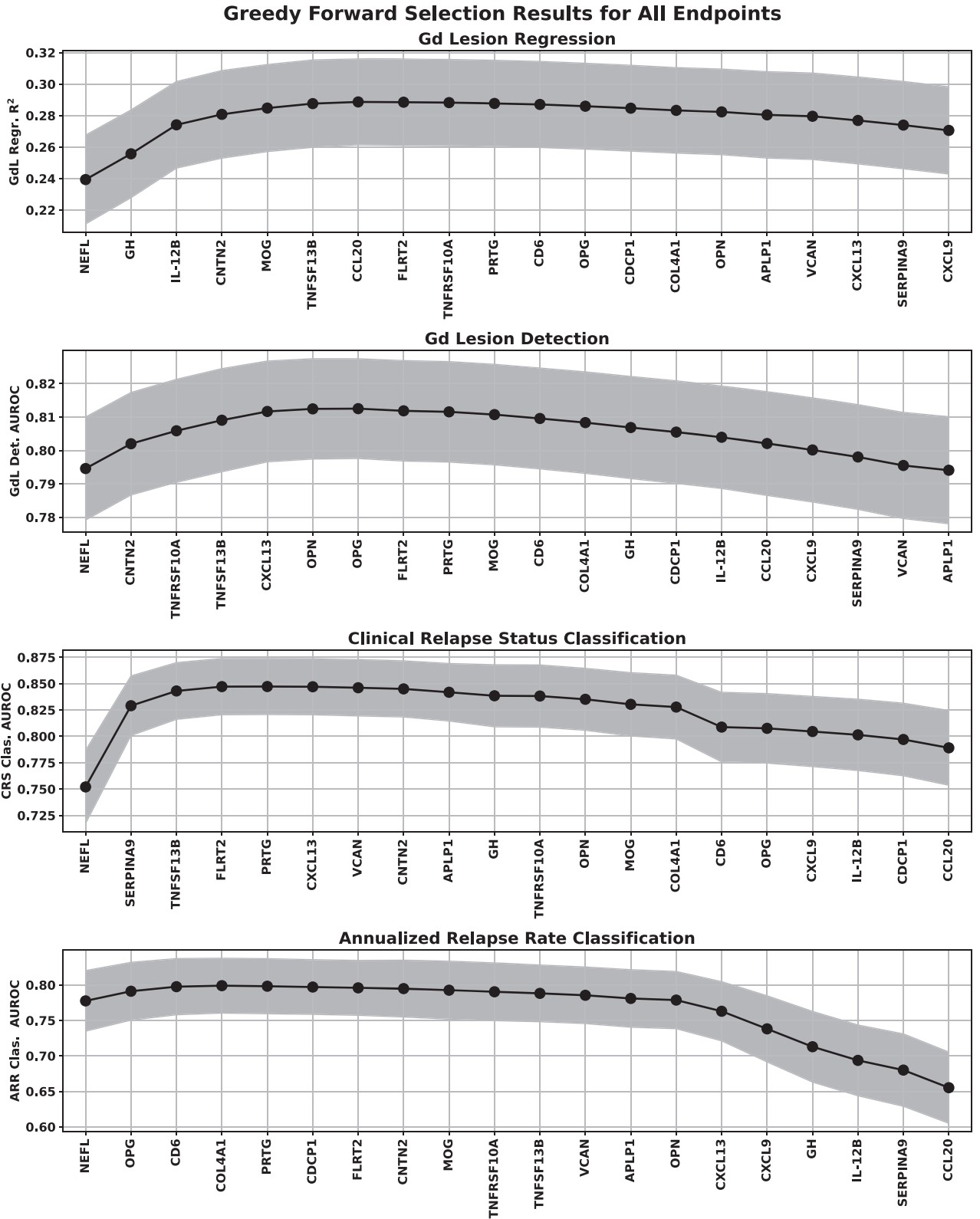

**Fig. 4 | GFS curve for the regression (top) and classification (bottom three) analysis of all endpoints.** The points represent the mean over the bootstrap splits and the shaded region represents the standard deviation. The protein features selected for each of the multivariate analyses were Gd lesion regression (NfL, GH, IL-12B, CNTN2, MOG, TNFSF13B), Gd lesion classification (NfL, CNTN2, TNFRSF10A, CXCL13, TNFSF13B), Clinical relapse status (NfL, SERPINA9, TNFSF13B, FLRT2), and annualized relapse rate (NfL, OPG, CD6). Note that the regression analysis was clipped at a lesion count of five (only 5.6% of our samples had more than five lesions, making any model behavior above that range unreliable). Source data are provided as a Source Data file. AUROC area under the receiver operator characteristic, ARR annualized relapse rate, CRS clinical relapse status, Gd gadolinium, GFS greedy forward selection, NfL neurofilament light chain.

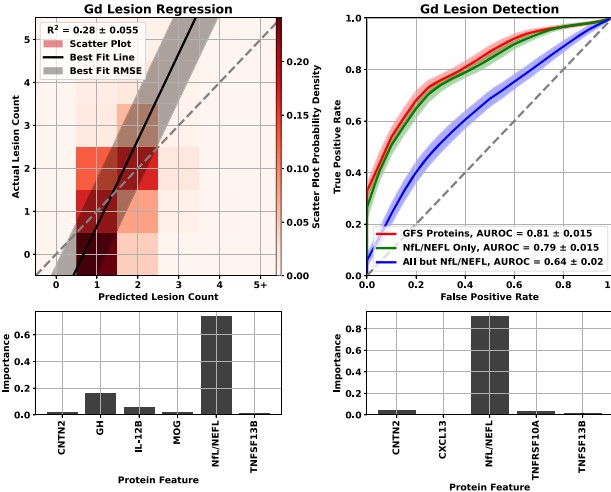

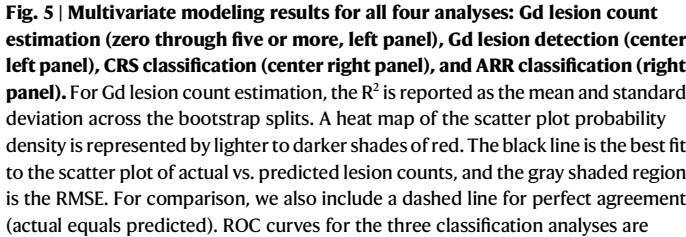

**Fig. 5 | Multivariate modeling results for all four analyses: Gd lesion count estimation (zero through five or more, left panel), Gd lesion detection (center left panel), CRS classification (center right panel), and ARR classification (right panel).** For Gd lesion count estimation, the $R^2$ is reported as the mean and standard deviation across the bootstrap splits. A heat map of the scatter plot probability density is represented by lighter to darker shades of red. The black line is the best fit to the scatter plot of actual vs. predicted lesion counts, and the gray shaded region is the RMSE. For comparison, we also include a dashed line for perfect agreement (actual equals predicted). ROC curves for the three classification analyses are represented as a solid line for the mean and shaded region for the standard deviation across all bootstrap splits using the following colors: red represents the model built with the greedy forward selection proteins, green represents the model built with NfL/NEFL only, and blue represents the model built with every protein but NfL/NEFL. Each analysis was plotted with the model performance plot above a feature importance bar graph for the GFS proteins. Source data are provided as a Source Data file. ARR annualized relapse rate, CRS clinical relapse status, Gd gadolinium, GFS greedy forward selection, $R^2$ square of Pearson's correlation coefficient.

Comparison of the actual versus estimated Gd lesion count (left panel, Fig. 5) showed that the regression model slightly overpredicts Gd lesion count for samples with no lesions and underpredicts it for those with five or more, while the model's accuracy for lesion counts of 1-2 anchored the performance of the model. While regression models performed acceptably, results from the classification analysis were considerably stronger. For all Gd lesion analyses, NfL had the largest effect (Table 3; Fig. 5). NfL performance as the strongest biomarker was further reflected in the feature importance plots at the two left bottom panels of Fig. 5. However, there is an incremental improvement in performance between NfL only and the GFS proteins, particularly in the Gd lesion regression analysis. Furthermore, the models using all proteins except NfL were significantly better than random chance. There is clearly signaling for this endpoint in the other proteins that is drowned out by the performance of NfL for predicting Gd lesions.

While NfL was the strongest performer for CRS and ARR as well, its lead over the other proteins was considerably attenuated. GFS protein performance for CRS outperformed all endpoints except EDA, and achieved similar ARR results to that of GDA. Multivariate separation of samples by CRS and ARR was significantly better than chance and significantly better than univariate NfL. Consistent with the Gd lesion analysis, multivariate models trained on all features except NfL displayed an AUROC significantly better than random chance.

## Biological context of protein-endpoint associations reveals heterogeneous pathophysiology and complex molecular crosstalk

The biological context of the 21 proteins listed in Table 1 was analyzed in the following two ways: direct spatial, functional, and gene expression correlations to a curated set of open-source databases, and expanded graph techniques using the Scalable Precision Medicine Open Knowledge Engine (SPOKE).

For the direct correlation analysis, protein concentration values were associated with data from the Human Protein Atlas[20] (aggregated lymphoid tissue, peripheral blood immune cell type, and brain region proteomic data) as well as the Allen Brain Atlas[21] (brain structure, cell type, and transcriptomic data). Protein-protein interaction modeling was performed by inputting proteins into STRING[22] for network construction. Physically and functionally associated proteins that exhibited a minimum interaction score of 0.7 (high confidence) were classified as interacting with each CAP protein. Markov Clustering[23,24] was used to detect distinct subgraphs of interconnected proteins. Topological surveillance and centrality metric calculations were performed in Cytoscape[25]. Enrichr[26] was leveraged to functionally annotate protein subgraphs.

Comparative proteomic analysis across human organs, tissues, and cell types helped home in on MS-relevant information by isolating organ-specific blood analytes to address the blood's pervasive nature. This revealed the MS-specific locality of CAP proteins, facilitating downstream directionality assessment and revealing a rich repertoire of cell types related to MS. Constitutive expression of these proteins helped to facilitate downstream mechanistic modeling. The resultant

**Table 2 | Classification and regression performance for all analyses of all endpoints for different feature sets[a]**

| Analysis | GFS proteins | NfL/NEFL only | All except NfL/NEFL |
|---|---|---|---|
| Gd lesion count regression ($R^2$) | 0.280 ± 0.027 | 0.233 ± 0.028 | 0.073 ± 0.018 |
| GDA (AUROC) | 0.813 ± 0.015 | 0.794 ± 0.016 | 0.636 ± 0.020 |
| SDA (AUROC) | 0.720 ± 0.024 | 0.698 ± 0.024 | 0.599 ± 0.026 |
| EDA (AUROC) | 0.920 ± 0.014 | 0.907 ± 0.015 | 0.700 ± 0.028 |
| Clinical relapse status (AUROC) | 0.845 ± 0.026 | 0.751 ± 0.035 | 0.658 ± 0.040 |
| Annualized relapse rate status (AUROC) | 0.803 ± 0.039 | 0.781 ± 0.042 | 0.544 ± 0.064 |

[a]We report mean performance and its uncertainty (standard deviation) across all bootstrap simulations.

*AUROC* area under the receiver operating characteristics curve, *EDA* extreme disease activity, *Gd* gadolinium, *GDA* General disease activity, *GFS* genome-based peptide fingerprint scanning, *NfL* neurofilament light chain, $R^2$ the square of Pearson's correlation coefficient, *SDA* Subtle disease activity.

**Table 3 | Demographic and clinical breakdown of all three patient cohorts**

| Clinical status | Age [y] (mean ± SD) | MS dis. dur. [y] (mean ± SD) | Female [%] | Subtype [%] (RR,SP,PP,CIS) | DMT Type [%] (DF,G,I,Na,No,Ot,S1) | Samples |
|---|---|---|---|---|---|---|
| **CLIMB** | | | | | | |
| Gd+ | 38.1 ± 9.4 | 7.1 ± 6.1 | 74 | 93.9, 5.7, 0.4, 0.0 | 3, 22, 16, 4, 40, 12, 2 | 228 |
| Gd− | 40.5 ± 8.0 | 8.5 ± 6.4 | 74 | 93.9, 6.1, 0.0, 0.0 | 2, 33, 9, 2, 37, 15, 2 | 98 |
| Low ARR | 39.7 ± 9.6 | 8.7 ± 7.1 | 73 | 96.6, 3.4, 0.0, 0.0 | 3, 28, 8, 3, 41, 13, 2 | 148 |
| High ARR | 31.8 ± 7.5 | 2.0 ± 1.2 | 65 | 76.9, 23.1, 0.0, 0.0 | 0, 8, 31, 8, 31, 15, 8 | 13 |
| All | 38.8 ± 9.1 | 7.7 ± 6.2 | 73 | 93.9, 5.8, 0.3, 0.0 | 3, 25, 14, 3, 39, 13, 2 | 326 |
| **EPIC** | | | | | | |
| Gd+ | 40.3 ± 9.1 | 9.4 ± 8.9 | 74 | 75.6, 22.2, 1.5, 0.7 | 0, 12, 27, 0, 43, 17, 0 | 135 |
| Gd− | 43.9 ± 10.4 | 11.6 ± 8.8 | 71 | 77.8, 20.0, 0.0, 2.2 | 0, 7, 20, 0, 22, 51, 0 | 45 |
| All | 41.2 ± 9.5 | 9.9 ± 8.9 | 73 | 76.1, 21.7, 1.1, 1.1 | 0, 11, 25, 0, 38, 26, 0 | 180 |
| **ACP** | | | | | | |
| Clin. Act. | 38.7 ± 10.1 | 1.2 ± 2.3 | 77 | All RRMS | Not Available | 60 |
| Clin. Inact. | 35.5 ± 9.3 | 3.8 ± 2.1 | 73 | All RRMS | Not Available | 64 |
| All | 38.8 ± 9.6 | 2.5 ± 2.5 | 75 | All RRMS | Not Available | 124 |

*ACP* Accelerated Cure Project, *ARR* Annualized relapse rate, *CLIMB* Comprehensive Longitudinal Investigation of MS at Brigham and Women's Hospital, *EPIC* Expression, Proteomics, Imaging, Clinical at UCSF, *Gd* Gadolinium, high ARR ≥ 1.0 relapses per year, low ARR ≤ 0.2 relapses per year, *MS* multiple sclerosis, *RR* Relapsing/Remitting, *SP* Secondary Progressive, *PP* Primary Progressive, *CIS* Clinically Isolated Syndrome, *DF* Dimethyl Fumarate, *G* Glatiramer, *I* Interferon, *Na* Natalizumab, *No* None, *Ot* Other, *S1* S1P.

mapping of the 21 CAP proteins onto 10 biological hallmarks of MS is presented in the left panel of Fig. 6. The 10 hallmarks were each subdivided into two to three related biological processes. CAP proteins were then sorted into these processes based on their correlations with the databases and tools mentioned.

The SPOKE Neighborhood Explorer (https://spoke.rbvi.ucsf.edu) allows targeted exploration of any component of the SPOKE knowledge graph. SPOKE is a comprehensive graph with millions of biomedical concepts and has been previously utilized for drug repurposing[27], to conduct genetic analyses[28], and for clinical predictions[29]. SPOKE has integrated data from close to 40 databases, including ChEMBL, OMIM, LINCS, and the Human Protein Atlas, among others. As such, SPOKE contains all human genes and proteins, more than a million pharmaceutical compounds, more than 7000 diseases, and a comprehensive representation of signaling and metabolic pathways. Further, biological interpretation of the 21 proteins also included spatial expression profiling, Protein-Protein interaction modeling, and Gene Set Enrichment. The SPOKE knowledge graph for the CAP proteins is shown in the right panel of Fig. 6.

## Discussion

We evaluated multivariate analyses of blood serum biomarkers from three independent cohorts and identified twenty proteins that were strongly associated with increased clinical and radiographic activity of MS. To quantify MS activity, we focused principally on the presence of Gd+ lesions compared with patients lacking such lesions. We also examined the following two clinical measures of MS relapse activity: clinician assessment of relapse state at or near the time of blood draw, and the ARRs in the time leading up to the draw.

Protein feature-selection processes can inherently introduce bias. To minimize bias, we deliberately attempted to balance weight and normalization strategies. For example, we weighted each feature importance by the AUROC of the model in which it appeared. When AUROC is calculated on the same data on which the model was trained, it tends to bias the result in favor of proteins that perform well in large models. Larger models are more prone to overfitting noise in the training data, which will lead to systematically higher AUROC values. Also, normalizing the vector of feature importance values for each model to 1 tends to favor proteins that perform better in models with only a few features. Since the total importance score for models with fewer features is split among fewer proteins compared with ones with larger feature sets, the values themselves will be numerically higher.

There are two points that strongly favor multivariate modeling over univariate NfL regarding the Gd endpoint. First, the optimized multivariate model significantly outperformed univariate NfL in every framing of the problem of using serum protein chemistry to predict Gd lesion activity. Second, we find statistically significant performance in a multivariate modeling even when the model ignores NfL concentrations. Both statements hold for an endpoint that should be most advantageous to the performance of univariate NfL (Gd lesions), and the difference was even larger for the clinical relapse endpoints. Furthermore, univariate NfL lacks specificity, as NfL levels are known to be elevated for neurodegenerative conditions other than MS[30].

NfL showed the strongest univariate correlation with the radiographic and clinical measures of DA examined in this study. Furthermore, multivariate techniques showed increased performance compared with NfL alone for all analyses of all endpoints. Similarly, multivariate models without NfL were still significantly more performant than chance for all endpoints, suggesting biological signal from the other CAP proteins as well. The inclusion of additional inflammatory and neurodegenerative protein biomarkers can provide deeper insights and reveal stronger correlations to clinical and radiographic DA than NfL individually. Cytokines, chemokines, and other immune-related molecules have consistently been associated with DA in patients with MS and they constitute an attractive target for interrogation in biological samples from patients at different stages of their disease course. Additionally, measuring protein concentration has several advantages over transcriptional profiling, including higher stability and more straightforward biological interpretation of the results. A biological pathway-centered approach using a subset of those shown in the left panel of Fig. 6 is likely to be a successful strategy for planning future investigation.

In addition to overall efficacy of the models used to examine the endpoints in this study, information from a broader panel of serum biomarkers allowed for insights into the pathophysiology of new Gd lesions and clinical relapse. This includes the identification of chemokines that are significantly altered by DA mediated by inflammation or immune response. A multi-protein panel like the one developed in this study has the capability to capture the state of a patient's MS from multiple angles, allowing for a fuller picture of their pathophysiology.

This study leveraged three large, well-characterized cohorts of patients with MS, and evaluated well over 1000 protein biomarkers using highly sensitive assays and applied network modeling to the findings to provide insights into MS. We used a systems biology

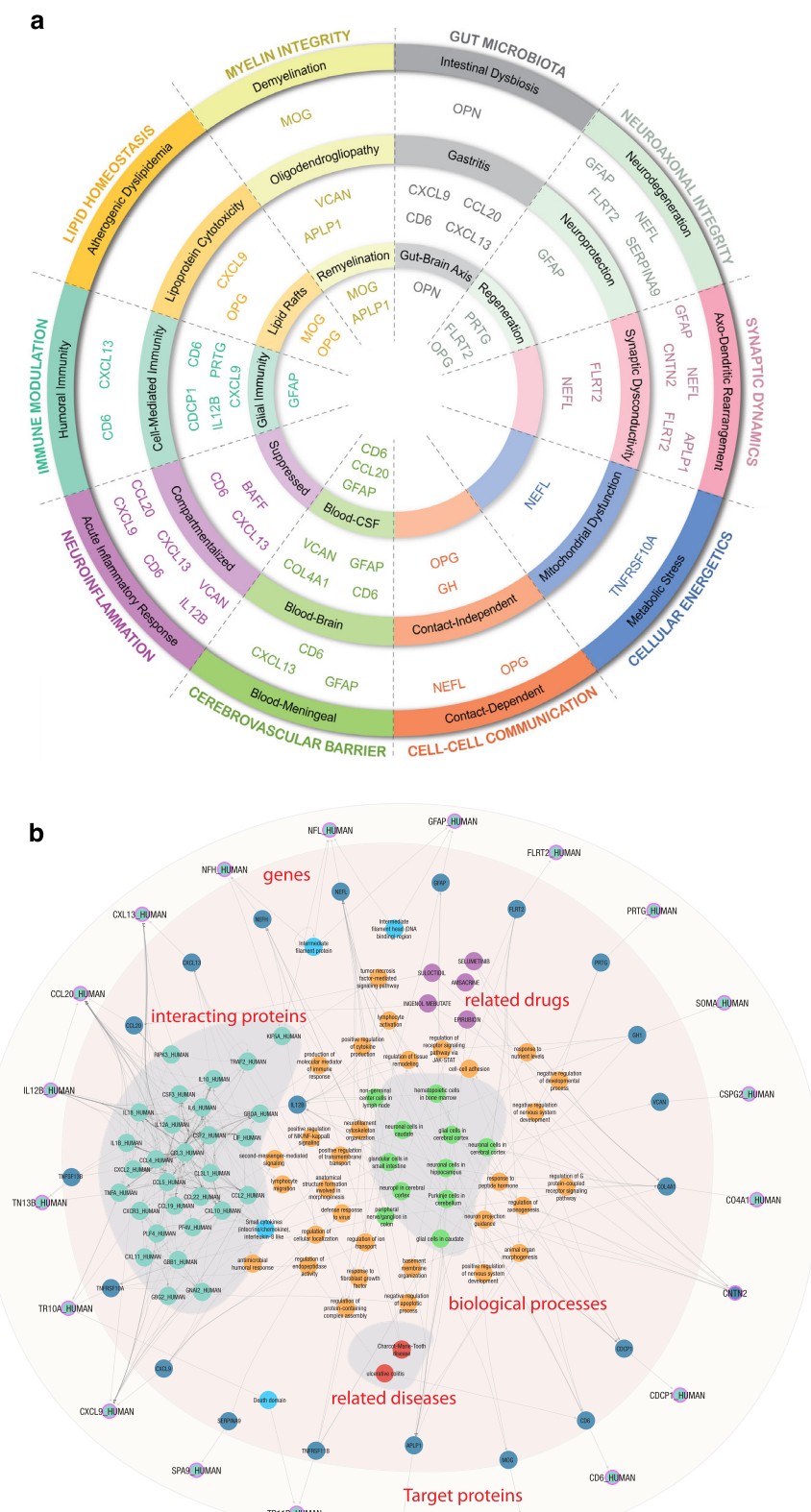

**Fig. 6 | CAP proteins sorted into biological processes and grouped into 10 MS hallmarks (a) and SPOKE graph visualization of biological neighborhood of CAP proteins (b). a** CAP proteins sorted into 10 MS hallmarks, categorically grouped by color, representing associated biological processes using each protein's correlation to spatial, functional, and gene expression data. **b** SPOKE graph visualization of biological neighborhood of CAP proteins. Using proteins as inputs (light blue circles with purple borders) results in a fully connected module including encoding genes (dark blue circles), directly interacting proteins (teal circles), their domains (sky blue circles), biological processes (orange circles) and a short list of related diseases (red circles). CAP custom assay panel, MS multiple sclerosis, SPOKE Scalable Precision Medicine Open Knowledge Engine.

framework to contextualize the mechanism-of-action of selected serum protein biomarkers with respect to MS DA. Through a complementary integration of machine learning and functional network analysis, we were able to shed light on the heterogeneous pathophysiological underpinnings of MS and unveil the orchestrated crosstalk between the various molecular facets of the disease. The analysis presented here was, however, purely cross-sectional. Large-scale longitudinal studies will be necessary to better understand MS and its evolution over time to unlock the potential of personalized MS care.

The end goal of the research program, the beginning of which is presented in this article, is a MS DA test that is fully validated in a clinical trial. Such a test would have tremendous clinical utility for many issues in MS care, including identification of active relapse, prediction of impending relapse, confirmation of NEDA status, assessment of patient-specific longitudinal changes relative to previous tests, and response to disease-modifying therapies (DMT). MS relapses can be quite subtle, especially early in the disease course, and can be easily confused with recurrences of symptoms in the setting of stressors (pseudo-flares) or conditions other than MS. This can lead to either the misattribution of an unrelated symptom to an MS relapse or to an early relapse being misattributed to some other clinical event. An inexpensive, clinically simple, precise, repeatable test for MS relapses would go a long way toward reducing both types of errors. Response to relapses often includes an escalation in DMT or steroid doses, both of which can have side effects. A test like the one we describe here has the potential to also serve as a leading indicator of impending relapses ahead of clinical presentation. This capability would need to be tested and validated clinically to quantify the scale of such a lead time. NEDA is the clinical gold standard for MS care. A NEDA designation indicates the patient's clinical and radiographic activity are held in check. A test like this one could alert a patient's MS care team to otherwise subclinical (or sub-radiographic) DA. An underlying truth beneath many scientific discoveries is that differential measurements tend to be simpler and more accurate than absolute ones. This is likely true of quantitative DA measurements as well. Some absolute level of DA score would be instructive to both patient and clinician, but deviations from baseline levels for an individual patient would be more so.

The ability to quantify the level of DA with a test like the one proposed could serve as an endpoint for clinical trials of DMT drugs and help clinicians to evaluate the efficacy of a DMT for a particular patient in less time than it would take for clinical or radiographic evidence to present. This would greatly enhance the quality of life and increase the health span of patients with MS and offer insight into patient adherence to their treatment plan.

Analytical validation of this DA panel will be followed by clinical validation studies to verify association with DA endpoints (primarily Gd lesions) in multiple independent cohorts[31]. Expansion of the test's clinical utility will be investigated with future studies to evaluate biomarker correlations with endpoints associated with MS DP, therapy selection, and differential diagnosis.

## Methods
### Ethics statement
The study protocol and study procedures were approved by institutional review boards and independent ethics committees at each study site. The University of California San Francisco Institutional Review Board granted ethical approval for the Expression, Proteomics, Imaging, Clinical at UCSF (EPIC) cohort. Mass General Brigham Human Research Committee granted ethical approval for the Comprehensive Longitudinal Investigation of MS at Brigham and Women's Hospital (CLIMB) cohort. Western Institutional Review Board, Copernicus Group IRB, Sheperd Center Research Review Committee, Institutional Review Board for Human Research at St. Joseph's Hospital and Medical Center, University of Massachusetts Medical School Committee for the Protection of Human Subjects in Research, Ohio State University

Biomedical Institutional Review Board, Beth Israel Deaconess Medical Center Committee on Clinical Investigations, Johns Hopkins Medicine Office of Human Subjects Research Institutional Review Board, and the Southwestern Medical Center Institutional Review Board all granted ethical approval for the Accelerated Cure Project (ACP) cohort.

### Materials
Serum samples were obtained from a subset of three deeply phenotyped cohorts were analyzed for protein levels and associated with clinical and radiographic endpoints and a subset of the associated clinical manifest to select features for inclusion in a custom assay panel and used in a subsequent cross-sectional analysis. The three endpoints in this study were: presence of Gd lesions for samples (defined as samples for which the blood draw was performed within 30 days of a contrast-enhancing magnetic resonance imaging, with the count of Gd lesions determined by a neuroradiologist), clinically active MS versus clinically inactive MS samples (defined as samples for which the blood draw was performed during a state of active relapse or inactive remission as defined by a clinician), and high versus low ARR samples (defined as those for which the blood draw was performed, and corresponding ARR-derived binary labels were determined (high: ≥1.0 and low: ≤ 0.2 relapses per year).

These three endpoints were taken from three different cohorts of patients and samples. All three cohorts are much larger than the subsets analyzed in this cross-sectional study. Samples were chosen to balance occupancy across the Gd lesion and CRS endpoints (ARR represented more of an opportunistic endpoint). The three cohorts contributing samples to this study were CLIMB (CLIMB endpoints: radiographically defined relapse status using Gd lesions [primary], ARR [secondary]), EPIC (EPIC endpoint: radiographically defined relapse status using Gd lesions), ACP (ACP endpoint: clinically defined relapse status - active versus inactive).

A total of 506 samples from CLIMB and EPIC were included in the Gd lesion analysis. One hundred and twenty-four samples from ACP contributed to the CRS endpoint. A subset of 161 of the CLIMB samples, for which ARR was available, were used for that analysis. A detailed summary of the salient demographic and clinical features from each cohort is presented in Table 3.

### Application of selected proteins to study endpoints
After selecting the top 20 performing proteins (20, not 21 because one spot in the panel was saved for a desired protein being added to the Olink platform–see Results for further discussion), we next evaluated their performance in the clinically relevant endpoints in an independent statistical analysis. The Gd lesion endpoint was investigated using both a classification and a regression approach. Specifically, a logistic regression model was trained to classify each serum sample as being associated with some positive number of Gd lesions or not, and a Poisson regression model was trained to estimate the number of Gd lesions associated with each sample. Investigation of the CRS involved the analysis of blood serum taken from patients during a clinically active relapse (exacerbation) or during a period of clinical stability, or inactive status (quiescence), following an approach similar to that taken for lesion presence classification. Finally, we addressed the ARR endpoint by considering the question of whether each sample had ARR < 0.2 relapses per year (low) or >1.0 relapse per year (high). Recognizing that the modest sample size and low positivity rate of ARR (6.9%, or 13 out of 188 samples) impact the power of this analysis, the same overall strategy as the two previous binary endpoints was used.

Due to the finite timescale on which brain lesions will be enhanced under Gd contrast, we precleaned our Gd lesion data by discarding all serum samples drawn more than 30 days from (before or after) the magnetic resonance imaging from which the lesion count was extracted. We accounted for batch-to-batch variability in relative quantitation by measuring bridge normalization samples across all

**Table 4 | Intra- and inter-assay percent coefficients of variation (%CV)**

| Biomarker | Intra-assay %CV | Inter-assay %CV |
|---|---|---|
| APLP1-Olink METABOLISM | 17 | 28 |
| CCL20-Olink INFLAMMATION | 11 | 16 |
| CD6-Olink INFLAMMATION | 10 | 12 |
| CDCP1-Olink INFLAMMATION | 11 | 11 |
| COL4A1-Olink CELL REGULATION | 18 | 16 |
| CNTN2-Olink ORGAN DAMAGE | 16 | 16 |
| CXCL13-Olink ONCOLOGY II | 11 | 16 |
| CXCL9-Olink INFLAMMATION | 9 | 11 |
| FLRT2-Olink NEUROLOGY | 12 | 20 |
| GH-Olink CARDIOVASCULAR II | 15 | 23 |
| IL-12B-Olink INFLAMMATION | 10 | 10 |
| MOG-Olink CELL REGULATION | 16 | 14 |
| NEFL-Olink NEURO EXPLORATORY | 8 | 17 |
| OPG-Olink INFLAMMATION | 9 | 11 |
| OPN-Olink CARDIOVASCULAR III | 17 | 22 |
| PRTG-Olink NEUROLOGY | 8 | 11 |
| SERPINA9-Olink ORGAN DAMAGE | 11 | 13 |
| TNFRSF10A-Olink CARDIOVASCULAR II | 11 | 10 |
| TNFSF13B-Olink CARDIOVASCULAR III | 14 | 18 |
| VCAN-Olink METABOLISM | 18 | 18 |

assays. Intra- and inter-assay percent coefficients of variation are reported in Table 4.

The univariate analysis of all three endpoints was performed using the bridge-normalized NPX values described above. The multivariate analysis for each endpoint used those same NPX values after correction for the following clinical variables: age, sex, and MS disease duration. An ordinary least squares linear regression model was fitted to the batch-normalized NPX values of the Gd-negative samples for each protein using the clinical variables as features. This estimate was then subtracted from all sample NPX values to make the demographic/clinical correction. The residual from this correction process was used as features in multivariate modeling.

Throughout each stage of the multivariate modeling process, an ensemble of bootstrap simulations was generated to control for overfitting. The data for each endpoint was randomly split into train (two-thirds) and test (one-third) subsets 1000 times. A different model of the same configuration was then trained on each bootstrap split, and model performance was taken to be the mean across the ensemble (with uncertainty parameterized by the square root of the variance). We chose bootstrap simulations instead of k-fold cross validation because the particular random split in the cross validation could have small effects on the outcome of some of the modeling efforts. The large number of random splits generated in the bootstrap process smoothed contributions from the small number of outlier samples in our data.

Both the regression and classification analyses were performed on an optimized subset of the entire panel chosen using GFS of protein features, with $R^2$ and AUROC, respectively, as the target metric and average over the ensemble of bootstrap splits. GFS starts with the top-performing univariate protein then checks each of the remaining features to construct the most performant two-protein model. With that in hand, each of the remaining proteins is checked in turn to see which combination provides the best three-protein performance. The process continues until all available proteins are included in the model. Optimal model size is then determined as the feature set size where performance is no longer significantly improved by adding an additional protein feature. In general, this function reaches a plateau value of optimal performance and then turns over as extra features are added that contribute only noise to the model. This is often a global maximum value of all possible combinations of features but was taken for the purposes of our study as the optimal model size and performance.

## Statistics and reproducibility

**Analytical methods.** For the purposes of the screening studies, up to 1411 proteins were measured using two separate immunoassay platforms. The first panel of 1196 proteins were analyzed using Proximity Extension Assay technology on the Olink™ Platform[32]. Protein concentrations were reported as NPX values (normalized protein expression), which provide expression levels relative to the other samples included on the plate and within the batch. An additional panel of 215 proteins were analyzed using xMAP® technology immunoassays at Myriad RBM, Inc. (RBM). Absolute protein concentrations from the RBM platform were determined using calibrated standard curves.

Exploratory data analysis was conducted to filter noise, reduce dimensionality, and avoid collinearity. Univariate significance was combined with multivariate importance from models generated with randomly selected combinations of different numbers of proteins to select features for inclusion into the custom assay panel. Biomarkers selected as features in the panel were investigated for relevance and interactions using biological network models. A 21-plex custom assay panel was then manufactured and analytically validated[33] to establish the following specifications and parameters: accuracy, precision, sensitivity, specificity, reference ranges, stability (reagents and samples), diurnal variation, drug interference, and assay robustness. The custom assay panel has been manufactured to include calibrators to report results in absolute concentration and a fit-for-purpose analytical validation has been performed. All samples previously run in the biomarker screening studies were reanalyzed using the custom assay panel.

**Biostatistical methods.** The program outlined in this report can be organized into the following two phases: the feature selection of the final 21 protein analytes and the studies that use those proteins as features to examine the three study endpoints. The former balanced information about all three endpoints to select a final ensemble of proteins for use as features in a more detailed cross-sectional analysis of all three endpoints in the latter phase. All analyses were performed in the python[34] programming language (version 3.11.7), making use of the SciPy[35] (version 1.11.4) and Scikit-Learn[36] (version 1.2.2) packages for statistical tests and machine learning models respectively.

**Protein feature selection.** The protein feature selection phase of this analysis preceded and was completely independent of the work reported in the Results section of this article. That analysis used only the 20 proteins chosen in the protein feature selection exercise described in this section. We followed two parallel tracks to identify the proteins most strongly associated with the three study endpoints.

First, we looked at the univariate correlation between each of 1411 proteins (1196 on the Olink platform and 215 from RBM). Gd lesions were treated as a binary variable used to classify samples based on their presence or absence (i.e., zero lesions vs. one or more). This allowed us to treat this endpoint consistently with the other two, which are inherently binary in nature. We computed the AUROC for separating the positive from negative samples for all three endpoints for all 1411 proteins and ranked them in decreasing order for each, paying special attention to analytes that showed a strong association with more than one endpoint.

To avoid biasing our multivariate feature selection process toward proteins that performed well in one but not another model architecture (e.g., tree-based vs. linear models), we investigated each of the following model types: logistic regression, support vector

classifiers, and random forest classifiers. Furthermore, we did not have an a priori estimate of the optimal number of features in each model, so we tested models with 3–21 proteins in steps of three. Because of its strong univariate performance in our data and its well-established association with many aspects of MS pathophysiology in the published literature, we used NfL (NfL is referred to by its gene name NEFL on the Olink platform, so we used the two names interchangeably in the analysis of these data) as the seed for multivariate feature selection analyses. We then randomly selected proteins *in addition to NfL* to fill out the set number of features (3–21, by threes) and used them to construct models of all three architecture types. This was repeated 100,000 times for each combination of model size and architecture. All models were trained on the entire dataset and evaluated for efficacy against the endpoint under consideration by calculating AUROC. After generating this ensemble of models, we extracted feature importance values from each one and tracked the feature importance for each protein over all the models in which it appeared.

In logistic regression models, feature importance was taken to be the variance-normalized absolute value of each feature coefficient. Protein concentration was expressed in NPX for this study, which is analogous to the log of the absolute concentration of each protein. The variance across all samples therefore did not numerically vary as widely across protein as it would if we were using linear concentration values. In support vector classifiers, we took feature importance to be the feature coefficient absolute value. We used a linear kernel for our support vector classifier models so that feature coefficients were well defined. In random forest classifier models, we simply used the Gini coefficients attached to the model.

Feature importance vectors were all normalized to unit sum for each model. When averaging the feature importance value for each protein across all models in which it appeared, we weighted each importance by the AUROC of its model so that proteins appearing in highly efficacious models would be favored over ones in models with poor performance. We then ranked each protein by its average AUROC-weighted feature importance so that we could quantify the multivariate performance of each protein separately from its univariate performance. We selected the top performers integrated across each endpoint's univariate and multivariate lists. The univariate and multivariate feature importance was then weighed along with more operational/analytical constraints (e.g., analyte precision and stability, confounding temporal or behavioral dependencies, association with biologically interesting physiological pathways) in the choice of the 20 proteins included in the assay. The final list of selected proteins is compiled in Table 1 of this report.

### Reporting summary
Further information on research design is available in the Nature Portfolio Reporting Summary linked to this article.

## Data availability
The data that support the findings of this study are openly available in GitHub at https://github.com/vmgehman/infl-neurodeg-bmkr-ms. Source data are provided with this paper.

## Code availability
The code that supports the findings of this study is openly available in GitHub at https://github.com/vmgehman/infl-neurodeg-bmkr-ms.

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

## Acknowledgements

Editorial assistance was provided by Maureen Wallace-Nadolski, PhD, of The Lockwood Group (Stamford, CT) and was supported by funding from Octave Bioscience, Inc (Menlo Park, CA).

## Author contributions

T.C., F.Q., V.M.G, M.B., R.B, B.A.C.C., R.G., S.L.H., R.G.H, A.K., H.L., A.P., S.J.C., A.S., N.S., S.S., H.W., H.Y., and S.E.B. were involved in the conception and design of the study, data acquisition, and data analysis. V.M.G., M.B., A.K., H.L., and S.E.B. conducted the statistical analysis. T.C., F.Q., V.M.G., M.B., and S.E.B. drafted the manuscript. T.C., F.Q., V.M.G, M.B., R.B, B.A.C.C., R.G., S.L.H., R.G.H, A.K., H.L., A.P., S.J.C., A.S., N.S., S.S., H.W., H.Y., and S.E.B. critically reviewed the manuscript and approved the submitted version.

## Competing interests

The authors declare the following competing interests: T.C. has received compensation for consulting from Biogen, Novartis Pharmaceuticals, Roche Genentech, and Sanofi Genzyme, and received research support from the National Institutes of Health, National MS Society, US Department of Defense, EMD Serono, I-Mab Biopharma, Novartis Pharmaceuticals, Octave Bioscience, Inc, Roche Genentech, and Tiziana Life Sciences. V.M.G., M.B., and A.K. were employees of Octave Bioscience, Inc at the time the study was completed. R.B. is funded by the NMSS Harry Weaver Award, NIH, DOD, NSF, as well as Biogen, Novartis, and Roche Genentech. She has received personal fees for consulting from Alexion, EMD Serono, Horizon, Janssen, Sanofi-Genzyme, and TG Therapeutics. B.A.C.C. has received personal compensation for consulting from Alexion, Atara, Autobahn, Avotres, Biogen, Boston Pharma, EMD Serono, Gossamer Bio, Hexal/Sandoz, Horizon, Immunic AG, Neuron23, Novartis, Sanofi, Siemens, and TG Therapeutics, and received research support from Genentech. S.L.H. currently serves on the scientific advisory board of Accure, Alector, and Annexon; board of directors of Neurona; and has previously consulted for BD, Moderna, and NGM Bio. Dr. Hauser also has received travel reimbursement and writing support from F. Hoffmann-La Roche and Novartis AG for anti-CD20 therapy-related meetings and presentations and is supported by grants from the NIH/NINDS (R35NS111644). R.G. H. has received fees for consultation from Roche/Genentech, Novartis, Neuron23, QIA Consulting, and research funding from Roche/Genentech and Atara. H.L. has received research support from the US Department of Defense and Octave Bioscience, Inc. A.P. is currently an employee of Moderna Therapeutics. F.Q. is an employee of Octave Bioscience, Inc. N.S. has received the Sylvia Lawry Physician Fellowship Award from the National MS Society. She has also received compensation for consulting from EMD Serono. H.W. has received research support from the Department of Defense, Genentech, Inc., National Institutes of Health, National Multiple Sclerosis Society, Novartis, and Sanofi Genzyme. He has received compensation for consulting from Genentech, Inc., IM Therapeutics, IMAB Biopharma, MedDay Pharmaceuticals, Tiziana Life Sciences, and vTv Therapeutics. S.E.B. is co-Founder of Mate Bioservices. H.Y., R.G., S.S., S.J.C., and A.S. declare no competing interests.
