## [Peer Review File · Nature Communications]

Inflammatory and neurodegenerative serum protein biomarkers increase sensitivity to detect clinical and radiographic disease activity in multiple sclerosisREVIEWER COMMENTS

Reviewer #1 (Remarks to the Author):

The manuscript provides a detailed study on potential biomarkers for inflammatory and neurodegenerative processes typical of Multiple Sclerosis (MS). The paper is structured around protein selection, univariate analysis, multivariate modeling, and biological interpretation using the SPOKE knowledge graph. Overall, this manuscript promises to be a meaningful addition to the literature on MS biomarkers. However, some minor revisions are required to ensure clarity, context-specificity, and overall coherence of the study.

INTRODUCTION

- The term "liquid biomarker" as used in the context of serum neurofilament light chain (sNfL) is ambiguous. Does this term refer to biomarkers found in body fluids (like blood or CSF) as opposed to tissue-based biomarkers?
- Concerning the elevation levels of sNfL during relapses, on one hand, it's stated that sNfL is "30-35% higher during relapses," while in another section, it's described as being "only marginally elevated." For accuracy and clarity, the authors should reconcile these statements to present a consistent picture of the sNfL elevation during relapses. If the term "marginally" is used to describe a 30-35% increase, a justification might be needed to contextualize this for the readers.

METHODS

- Whilst the use of bridge normalization samples provides an approach to address batch-to-batch variability, it remains essential to report the coefficient of intra-assay and inter-assay variation.
- The manuscript mentions the selection of top performers across "each of our six lists." However, the exact nature and derivation of these six distinct lists remain ambiguous. The methodology described involves three machine learning techniques (logistic regression, support vector classifiers, and random forest classifiers) and feature selection across different model sizes (3-21 proteins in steps of three). It would be beneficial for clarity if the authors could provide a more explicit breakdown of how these combinations produce the mentioned six lists and what each list represents.
- A pivotal aspect of the study involves determining a set of proteins that have predictive power across all three binary outcomes. While the manuscript details the process of univariate and multivariate analyses, it remains less clear how the authors assured that the final selected proteins didn't overly favor one outcome over the others. For instance:
 - o From the univariate analysis, it's unclear how individual protein-outcome associations were carried forward or incorporated into the multivariate modeling phase.
 - o While the multivariate analysis provides an averaged, AUROC-weighted feature importance, it would be helpful to understand how this amalgamation ensures that the selected proteins aren't overly biased towards one specific outcome.
 - o The authors might consider providing additional clarity or validation results that show the selected proteins' efficacy across all three outcomes, ensuring that one outcome isn't overly favored at the

expense of the others.

- Authors adjusted for clinical variables (age, sex, MS disease duration) when conducting the multivariate analysis. However, was there a reason other potential confounders weren't considered? For instance, treatment status or other comorbidities.

- The authors should clearly state the threshold they used to determine statistical significance. The authors conducted univariate analyses on more than 1400 proteins for three outcomes. Given the sheer number of tests performed, was there a consideration to apply a multiple comparison correction, such as Bonferroni, to control for the inflated familywise error rate?

RESULTS

- The study utilizes various metrics, like ROC curves, RMS error, and AUROC, which are visualized in the figures. However, for the reader's convenience, it would be beneficial to include key numeric values of these metrics within the main text, particularly those that demonstrate the model's performance on the validation or test set.

Reviewer #2 (Remarks to the Author):

Interesting paper, with a good-sized cohort of serum samples from people with MS. Certainly, there is a need to find additional markers other than NfL that may offer pathophysiologic insight.

In table 3, it seems there are some age differences between groups. Of course, it is known that GAD positive lesions correlate with younger age. I wonder if mixed models or other adjustments were done for age (and other demographics) and biomarkers. For example, NFL is also known to have a z-score and can be corrected by BMI.

An essential piece of information missing is the MS phenotype. GAD lesions are more often seen in RRMS, but can be seen in SPMS or PPMS. This is not trivial as the immunologic and neurodegenerative milieu could be very different between phenotypes.

In figure 1, there is an intriguing trend in some markers, where there is no linear response. For example, NFL shows a steady increase among groups, with higher levels in higher GAD number categories. But other markers have more inverted "U" shapes, with higher levels in 1 GAD lesion and then lower in 3 or more GAD lesions (like VCAN, FLRT2, COL4A1), even in some included in the multivariate model (IL-12B, TNFSF13B). Sample numbers could be a big part, but this is odd when extracting biological interpretation. Why would growth hormone increase if you have 1 GAD lesion but not if you have 3 or more?

There are also intriguing findings in the correlations with relapses and ARR. For example, NFL is clearly higher in exacerbations and in high ARR. This makes sense, as one assumes a relapse is associated with

new acute neuroaxonal injury and NFL shedding, and also that someone with several relapses will have continuously high NFL. By the way, it is also unclear why GAD lesions, relapses and high ARR would be independent variables. One absolutely does see asymptomatic GAD lesions, but, a “true” relapse more often than not would be correlated with a new lesion, somewhere in the neuraxis (could be spinal, that this study does not clearly account for), that enhances, and is responsible for the relapsing symptoms. For example, a relapse of optic neuritis will show optic nerve enhancement, and so on.

So, there are some findings like for VCAN, PRTG, FLRT2, where they seem to be elevated in a relapse, but low in those with higher ARR (even if not all statistically significant). This is also counterintuitive. Furthermore, one would hardly say any ARR above 1 is “high”. Certainly, less than 0.2 is low, but in the pre DMT era, an ARR of 1 would be considered very moderate. Are there any other cut-offs? In keeping with this, DMTs are a major issue to consider. If there is measurement of immune markers, these could be drastically modified by immune modulators and steroids. I see no accounting for DMT, unless all samples came from untreated individuals? If not, the other issue is that DMT use is certainly related to chances of finding disease activity. This needs to be accounted for.

In the interpretation there are comments on the specificity of the findings. One would want to see a non MS inflammatory control cohort (rheumatoid?) or non MS CNS inflammatory control (infection?), or perhaps even other neurological disease control (stroke? Dementia?).

There are no correlations with clinical measures of disability such as EDSS. This should be available in deeply phenotyped samples. Issues like comorbidities, BMI, smoking, etc. could also be accounted for.

In the ROC curves, it is true the combination of markers with NFL gives a statistically significant improvement in discrimination, but the question is if the marginal absolute difference would be cost effective. I would argue that the results strengthen the case of NFL as a single, clinically relevant marker, as opposed to an interpretation of “personalized” biomarkers.

The soundness of complex and sequential statistical methods could be reviewed by an independent statistician, if thought needed. by the editors.

Reviewer #3 (Remarks to the Author):

Inflammatory and neurodegenerative serum protein biomarkers increase sensitivity to detect disease activity in multiple sclerosis.

The Authors reported the results from a cross-sectional study, using three cohorts of MS patients, aimed to unravel the prognostic ability of inflammatory and neurodegenerative serum protein biomarkers.

The first concern here is that the study design was not reported neither described. I supposed that serum was collected at the enrolment visit as well as the endpoints. But I’m not sure. I suggest to better describe the study design from an epidemiological perspective.

The three cohorts should be referenced both in (introduction/results and in the methods sections). Moreover, the CLIMB study is longitudinal (probably the EPIC and ACP studies too) therefore why did the Authors not use the prospective data (endpoints) to potentially elevate the interesting results from their study. Results from a prospective study can give answer on the causal relationships between protein biomarkers and can allow the Authors to deal with prediction and prognosis. On the contrary, the cross-sectional study, as reported here, can result only in associations (causal relationship is not guaranteed) and classification.

From a Statistical point of view, I found the methods section well written. The statistical tools were robust and sophisticated, but I have a major concern.

The relapse rate endpoint (number of relapse in a time window) should be analysed using a negative binomial model with the time as offset. These Authors used linear regression analysis. Moreover, they clipped the number of relapses at five which is not necessary. The last two methodological points could lead to, potentially, unreliable results.

In discussion section, page 12, lines 236-261, the authors reported the remarkable goals of their projects. The first step toward the achievement of these goals is the design and the conduction of a prospective study.

Specific comments:

Page 5, lines: 71-75. Despite the Authors refer to methods section, I suggest to report very briefly how they have selected 20 proteins from the original 1411 ones.

Page 5, lines 81-82. The Authors should report here and in the figure 2 legend how “low” and “high” annualized relapse rate were defined (not only in the methods section).

Page 5, lines 83-95. The Authors performed a Pearson correlation analysis which requires normal distribution and continuous nature of the two variables. Therefore, I did not agree to clip at five the number of lesions. In this analysis, Authors should use the unclipped number of lesions. Anyway, as I wrote before, this is not the appropriate analysis for the ARR endpoint.

Pag 18, line 394. R2 is not the Pearson Correlation coefficient. R2 is the square of the Pearson Correlation coefficient.

REVIEWER COMMENTS

Reviewer 1

The manuscript provides a detailed study on potential biomarkers for inflammatory and neurodegenerative processes typical of Multiple Sclerosis (MS). The paper is structured around protein selection, univariate analysis, multivariate modeling, and biological interpretation using the SPOKE knowledge graph. Overall, this manuscript promises to be a meaningful addition to the literature on MS biomarkers. However, some minor revisions are required to ensure clarity, context-specificity, and overall coherence of the study.

1. The term "liquid biomarker" as used in the context of serum neurofilament light chain (sNfL) is ambiguous. Does this term refer to biomarkers found in body fluids (like blood or CSF) as opposed to tissue-based biomarkers?

Author response and action taken: Our use of the term "liquid" refers to biomarkers found in body fluids (like blood or cerebrospinal fluid) as opposed to tissue-based biomarkers. We are placing this work in context of the broader literature focused on neurofilament light chain (NfL) in blood serum. We have reworded the text to make that clearer, including dropping the use of the word "liquid" (page 3, lines 50–51). The sentence now reads:

"Most studies have focused primarily on neurofilament light chain in blood serum (sNfL) as a biomarker in MS."

2. Concerning the elevation levels of sNfL during relapses, on one hand, it's stated that sNfL is "30-35% higher during relapses," while in another section, it's described as being "only marginally elevated." For accuracy and clarity, the authors should reconcile these statements to present a consistent picture of the sNfL elevation during relapses. If the term "marginally" is used to describe a 30-35% increase, a justification might be needed to contextualize this for the readers.

Author response and action taken: Thank you for your feedback on this word choice. We agree that the word "marginally" is subjective in its interpretation. We have changed this passage in the article to quantitatively place this neurofilament light chain in blood serum (sNfL) concentration increase in context with the group-level coefficient of variation in healthy individuals with a citation to the study measuring that value (Rosso M, et al. *Ann Clin Transl Neurol* 7, 945-955 [2020]). The passage has been revised to the following (page 3, lines 54–57):

"In the 3 months after a Gd+ lesion, an average 35% elevation in sNfL ($p < 0.0001$) was reported when compared with samples from patients in remission¹². Similarly, an average 32.3% elevation in sNfL was observed at the time of, or prior to a Gd+ lesion ($p = 0.002$) versus remission¹²."

3. Whilst the use of bridge normalization samples provides an approach to address batch-to-batch variability, it remains essential to report the coefficient of intra-assay and inter-assay variation.

Author response and action taken: We agree that the intra- and inter-assay coefficients of variation are important. These, and other assay performance criteria, are discussed at length in Qureshi F, et al. *Proteomics Clin Appl*, e2200018 (2023) in the context of the custom assay panel that was built in parallel with the analysis that went into this article. Furthermore, we have reported the intra- and inter-assay percent coefficient of variation for this study in a new table (**Table 4**).

Table 4 | Intra- and inter-assay percent coefficients of variation (%CV)

Biomarker	Intra-assay %CV	Inter-assay %CV
APLP1-Olink METABOLISM	17	28
CCL20-Olink INFLAMMATION	11	16
CD6-Olink INFLAMMATION	10	12
CDCP1-Olink INFLAMMATION	11	11
COL4A1-Olink CELL REGULATION	18	16
CNTN2-Olink ORGAN DAMAGE	16	16
CXCL13-Olink ONCOLOGY II	11	16
CXCL9-Olink INFLAMMATION	9	11
FLRT2-Olink NEUROLOGY	12	20
GH-Olink CARDIOVASCULAR II	15	23
IL-12B-Olink INFLAMMATION	10	10
MOG-Olink CELL REGULATION	16	14
NEFL-Olink NEURO EXPLORATORY	8	17
OPG-Olink INFLAMMATION	9	11
OPN-Olink CARDIOVASCULAR III	17	22
PRTG-Olink NEUROLOGY	8	11
SERPINA9-Olink ORGAN DAMAGE	11	13
TNFRSF10A-Olink CARDIOVASCULAR II	11	10
TNFSF13B-Olink CARDIOVASCULAR III	14	18
VCAN-Olink METABOLISM	18	18

4. The manuscript mentions the selection of top performers across "each of our six lists." However, the exact nature and derivation of these six distinct lists remain ambiguous. The methodology described involves three machine learning techniques (logistic regression, support vector classifiers, and random forest classifiers) and feature selection across different model sizes (3-21 proteins in steps of three). It would be beneficial for clarity if the authors could provide a more explicit breakdown of how these combinations produce the mentioned six lists and what each list represents.

- A pivotal aspect of the study involves determining a set of proteins that have predictive power across all three binary outcomes. While the manuscript details the process of univariate and multivariate analyses, it remains less clear how the authors assured that the final selected proteins didn't overly favor one outcome over the others. For instance:
 - o From the univariate analysis, it's unclear how individual protein-outcome associations were carried forward or incorporated into the multivariate modeling phase.
 - o While the multivariate analysis provides an averaged, AUROC-weighted feature importance, it would be helpful to understand how this amalgamation ensures that the selected proteins aren't overly biased towards one specific outcome.
 - o The authors might consider providing additional clarity or validation results that show the selected proteins' efficacy across all three outcomes, ensuring that one outcome isn't overly favored at the expense of the others.
- Authors adjusted for clinical variables (age, sex, MS disease duration) when conducting the multivariate analysis. However, was there a reason other potential confounders weren't considered? For instance, treatment status or other comorbidities.
- The authors should clearly state the threshold they used to determine statistical significance. The authors conducted univariate analyses on more than 1400 proteins for three outcomes. Given the sheer number of tests performed, was there a consideration to apply a multiple comparison correction, such as Bonferroni, to control for the inflated familywise error rate?

Author response and action taken: The focus of this report is not the protein feature selection. This section of the paper is background information, describing that part of the assay development process. The creation of a custom assay panel requires its developers to balance numerous factors that are often in tension with one another. These include, but are not limited to:

1. the statistical information contained in any particular protein signal for both univariate and multivariate analyses,
2. the performance of the analytes used to measure the concentration, both in terms of *de novo* accuracy, precision, and stability,
3. systematic physiological offsets associated with serum protein concentration (e.g. diurnal or ultradian variations),
4. connection between a particular protein concentration and one or more physiological/biological pathways,
5. the number of proteins available to include on a multiplexed assay panel.

With these numerous, nonorthogonal constraints in mind, there is no single figure of merit we can cite in our protein feature selection process for us to threshold and say, “proteins above this line are in the assay, and those below it are not”. The performance of individual or groups of proteins in the exploratory phase of the analysis described in this section of our article was one important factor, but far from the only concern when we constructed the list for our custom panel.

It also bears emphasizing that the analyses described in the “Protein feature selection” section were completely independent of those that yielded the actual results presented here (biostatistical methods are discussed in the “Application of selected proteins to study endpoints” section). The univariate and multivariate results presented in the sections bearing those names only included the 20 custom assay panel (CAP) proteins. Those and the biological interpretation discussed subsequently represent the real results in this article. To this end, we have rewritten parts of the “Protein feature selection” section to reflect its independence from the results of this report (pages 4–5, lines 88–92).

5. The study utilizes various metrics, like ROC curves, RMS error, and AUROC, which are visualized in the figures. However, for the reader's convenience, it would be beneficial to include key numeric values of these metrics within the main text, particularly those that demonstrate the model's performance on the validation or test set.

Author response and action taken: This is an excellent suggestion, doubly so given the likelihood of this manuscript being typeset with the figures and tables at the end of the paper instead of in-line with the text. We have included performance metrics for the models built from the CAP proteins selected by the greedy forward selection technique for all three endpoints (page 7, lines 153–158) in the text of the “Multivariate results” section (as we feel these represent the key findings of this analysis) and refer the reader to **Table 2** for details on the rest of the model space.

Reviewer 2

Interesting paper, with a good-sized cohort of serum samples from people with MS. Certainly, there is a need to find additional markers other than Nfl that may offer pathophysiologic insight.

1. In table 3, it seems there are some age differences between groups. Of course, it is known that GAD positive lesions correlate with younger age. I wonder if mixed models or other adjustments were done for age (and other demographics) and biomarkers. For example, NFL is also known to have a z-score and can be corrected by BMI.

Author response and action taken: We agree that it would have been very interesting to correct for body mass index and as well as a number of other demographic and clinical variables. We chose certain variables because they were available for all of the samples across all of the studies in our data set, and one of our goals was to treat all endpoints in a methodologically consistent manner so as to avoid a systematic difference between samples where we had different demographic and clinical data.

2. An essential piece of information missing is the MS phenotype. GAD lesions are more often seen in RRMS, but can be seen in SPMS or PPMS. This is not trivial as the immunologic and neurodegenerative milieu could be very different between phenotypes.

Author response and action taken:

We agree that MS phenotype is often featured and have included the MS phenotype breakdown for each segment of our cohort (positive and negative samples for each of the three endpoints) in Table 3. As you can see, the cohort is well balanced across MS subtypes for the Gd lesion and clinical status endpoints. Binarized ARR appears to be imbalanced in that there are too few RRMS and too many SPMS samples in the high ARR population. The difference is much less when you consider that there are only 13 high ARR samples. 10 (76.9%) of these were RRMS patients. If RRMS patients were represented at the same fraction as the low ARR cohort (93.9%), we would have seen just over 12 samples. Even this endpoint was balanced in the cohort to within Poisson counting fluctuations. Since our study was quite well-balanced with regard to MS subtype, we did not correct for it.

3. In figure 1, there is an intriguing trend in some markers, where there is no linear response. For example, NFL shows a steady increase among groups, with higher levels in higher GAD number categories. But other markers have more inverted “U” shapes, with higher levels in 1 GAD lesion and then lower in 3 or more GAD lesions (like VCAN, FLRT2, COL4A1), even in some included in the multivariate model (IL-12B, TNFSF13B). Sample numbers could be a big part, but this is odd when extracting biological interpretation. Why would growth hormone increase if you have 1 GAD lesion but not if you have 3 or more?

Author response and action taken: While it is quite clear that the only two proteins with a strong linear response are NEFL and MOG (CDCP1 and CXCL9 appear to just cross the line into significance, especially coupled with the fact that they show a Student’s *t* test statistic for general disease activity with the same sign; see Fig. 2), it is challenging to make a conclusive statement about an “inverted U” shape for any of the proteins you call out in Fig. 1. It appears that none of the proteins called out in this comment show an enhancement or suppression of concentration in any of the lesion bins in Fig. 1 above small fluctuations in the median that are well within the interquartile range (IQR). The only exceptions are APLP1 and VCAN, which show a small but significant increase from the median for zero lesions to any of the positive lesion bins, as seen in their Student’s *t* test coefficients in Fig. 2.

It is possible that a larger study (specifically, higher N in each of the lesion count bins) would have shown a clearer excess or deficit in one or more of the bins at population level in Fig. 1, but the IQR for every protein (even NFL/NEFL) is large enough that one must necessarily resort to multivariate modeling in order to make reliable predictions about the disease activity of any individual as well as a population.

4. There are also intriguing findings in the correlations with relapses and ARR. For example, NFL is clearly higher in exacerbations and in high ARR. This makes sense, as one assumes a relapse is associated with new acute neuroaxonal injury and NFL shedding, and also that someone with several relapses will have continuously high NFL. By the way, it is also unclear why GAD lesions, relapses and high ARR would be independent variables. One absolutely does see asymptomatic GAD lesions, but, a “true” relapse more often than not would be correlated with a new lesion, somewhere in the neuraxis (could be spinal, that this study does not clearly account for), that enhances, and is responsible for the

relapsing symptoms. For example, a relapse of optic neuritis will show optic nerve enhancement, and so on.

Author response and action taken: Excellent point. Part of our aim in treating the clinical and radiographic endpoints independently in this work was to address, not only the cases where an otherwise Gd-enhancing lesion occurs in the spine (which can be missed in a brain magnetic resonance imaging [MRI]), or is too small to be seen at the resolution of the MRI (many MRI scanners still operate at 1.5T rather than 3T, further exacerbating this problem), but also those where an MRI comes too late after the actual relapse for Gd contrast to work, if indeed they get scanned at all. If the blood-brain barrier rupture is repaired (or at least repaired enough to slow the bleed of Gd-loaded dye into the brain), then Gd-enhancing lesions would be nonexistent. This repair process can take as little as a few weeks, which can occur before many patients are scanned owing to lifestyle, socioeconomic, geographical, or other considerations over which patients and clinicians often have little control.

5. So, there are some findings like for VCAN, PRTG, FLRT2, where they seem to be elevated in a relapse, but low in those with higher ARR (even if not all statistically significant). This is also counterintuitive. Furthermore, one would hardly say any ARR above 1 is “high”. Certainly, less than 0.2 is low, but in the pre DMT era, an ARR of 1 would be considered very moderate. Are there any other cut-offs? In keeping with this, DMTs are a major issue to consider. If there is measurement of immune markers, these could be drastically modified by immune modulators and steroids. I see no accounting for DMT, unless all samples came from untreated individuals? If not, the other issue is that DMT use is certainly related to chances of finding disease activity. This needs to be accounted for.

Author response and action taken: We agree that disease-modifying therapy (DMT) status is an important systematic variable, and have added DMT status for each of our studies/endpoints in the cohort description table (Table 3 in the text). We did not, however, have access to that information for enough of our samples in this study to control for it. All of our samples are from well into the “post-DMT era”. While a patient with well-managed MS, including an efficacious DMT, should indeed appear lower in all three endpoints considered here, the reviewer is correct in noting that there could be fingerprints of different DMTs in the proteins measured for this study. An investigation of this relationship is rather outside the scope of our capabilities for this manuscript, but was considered in the clinical validation paper for the Multiple Sclerosis Disease Activity test that grew out of this work (Chitnis T, et al. *Clin Immunol* 253, 109688 [2023]).

6. In the interpretation there are comments on the specificity of the findings. One would want to see a non MS inflammatory control cohort (rheumatoid?) or non MS CNS inflammatory control (infection?), or perhaps even other neurological disease control (stroke? Dementia?).

Author response and action taken: This study is focused on the development of a disease activity assay to be deployed after the official diagnosis of MS, so a specificity test like the one you suggest would be outside the scope of work here. That said, you make another excellent suggestion for a future study

involving the development of a more diagnostic test. Such an investigation would be crucial to that effort and has long been on our list of things we would like to do as we collect more data from more patients. The closest we came to this was a number of serum pools from MS, inflammatory disease (rheumatoid arthritis), and healthy/normal patients we used as process controls in the assay. Unfortunately, serum pools are not really appropriate to include in analyses like this one. We would need to launch a new prospective study to tackle this properly.

7. There are no correlations with clinical measures of disability such as EDSS. This should be available in deeply phenotyped samples. Issues like comorbidities, BMI, smoking, etc. could also be accounted for.

Author response and action taken: As with the other clinical controls suggested by all three reviewers, we did not have Expanded Disability Status Scale (EDSS) scores for every one of our samples, so it was impossible to use it for model building in conjunction with all samples for all endpoints. That said, EDSS is an extremely important endpoint for MS disease progression (particularly from an explicitly mobility-based standpoint). As such, it will figure prominently in future work on quantifying the progression of MS.

8. In the ROC curves, it is true the combination of markers with NFL gives a statistically significant improvement in discrimination, but the question is if the marginal absolute difference would be cost effective. I would argue that the results strengthen the case of NFL as a single, clinically relevant marker, as opposed to an interpretation of “personalized” biomarkers.

Author response and action taken: A main focus of this paper is to examine the ability of a multi-protein assay to out-perform univariate sNfL in quantifying disease activity, both in terms of raw discrimination power across several endpoints, as well as with respect to accessing information about the underlying biology affecting a patient's MS status. We have clearly demonstrated statistically significant information in many other protein signals in addition to NfL. The cost-effectiveness of this test versus the use of sNfL concentration as a single biomarker is a commercialization problem and therefore outside the scope of a scientific article like this, but will be addressed in future studies.

9. The soundness of complex and sequential statistical methods could be reviewed by an independent statistician, if thought needed. by the editors.

Author response and action taken: We will happily submit our analysis to such a review if the editors deem it necessary.

Reviewer 3

Inflammatory and neurodegenerative serum protein biomarkers increase sensitivity to detect disease activity in multiple sclerosis.

The Authors reported the results from a cross-sectional study, using three cohorts of MS patients, aimed to unravel the prognostic ability of inflammatory and neurodegenerative serum protein biomarkers.

1. The first concern here is that the study design was not reported neither described. I supposed that serum was collected at the enrolment visit as well as the endpoints. But I'm not sure. I suggest to better describe the study design from an epidemiological perspective.

The three cohorts should be referenced both in (introduction/results and in the methods sections). Moreover, the CLIMB study is longitudinal (probably the EPIC and ACP studies too) therefore why did the Authors not use the prospective data (endpoints) to potentially elevate the interesting results from their study. Results from a prospective study can give answer on the causal relationships between protein biomarkers and can allow the Authors to deal with prediction and prognosis. On the contrary, the cross-sectional study, as reported here, can result only in associations (causal relationship is not guaranteed) and classification.

Author response and action taken: It perhaps bears reclarification here that the collaboration performing the analysis for this article was not the totality of the CLIMB, EPIC, or ACP cohorts. We certainly have a subset of members overlapping two of those three studies, but we did not have access to the entirety of any of those three studies. We obtained access to a subset of samples and clinical data from each of those studies in their capacity as a sample bank. Our ensemble of samples was intended to cover positive and negative cases of the three endpoints set out in the "Materials" section (where we also explicitly stated the inclusion criteria for said samples). We wholeheartedly agree that a prospective study would better elucidate underlying causal relationships. In fact, this is precisely what we did in designing the clinical validation study of the lab-developed test that grew out of this exploratory cross-sectional analysis (Chitnis T, et al. *Clin Immunol* 253, 109688 [2023]).

2. From a Statistical point of view, I found the methods section well written. The statistical tools were robust and sophisticated, but I have a major concern.

The relapse rate endpoint (number of relapse in a time window) should be analysed using a negative binomial model with the time as offset. These Authors used linear regression analysis. Moreover, they clipped the number of relapses at five which is not necessary. The last two methodological points could lead to, potentially, unreliable results.

Author response and action taken: This comment caused a fair amount of discussion on our part because this was not the way that we performed the annualized relapse rate (ARR) analysis. We binned clinician-determined ARR values at the time of blood draw into low (<0.2 relapses/year) and high

(>1.0/year) categories, then performed a classification analysis using a *logistic* regression to separate those two populations. If we had performed a regression analysis to predict ARR based on protein concentrations (as well as some set of features extracted from clinical history, most likely), a negative binomial regression or similar analysis would have indeed been appropriate. The only *linear* regression performed in this article was on the Gd lesion count endpoint, which seemed eminently appropriate for that problem. There was clearly a misunderstanding of the shape of our analysis here, and we have reworked the description of the ARR endpoint (page 18, lines 402–406) to make it clearer in the hopes that we avoid any further such issues.

3. In discussion section, page 12, lines 236-261, the authors reported the remarkable goals of their projects. The first step toward the achievement of these goals is the design and the conduction of a prospective study.

Author response and action taken: We heartily agree, and again refer the reviewer to our clinical validation paper mentioned above (Chitnis T, et al. *Clin Immunol* 253, 109688 [2023]). We have now included this citation in the manuscript (page 13, lines 289). We have also begun a number of other prospective studies with several collaborators as we continue this research and development program.

4. Page 5, lines: 71-75. Despite the Authors refer to methods section, I suggest to report very briefly how they have selected 20 proteins from the original 1411 ones.

Author response and action taken: Good idea. This suggestion has been incorporated (page 5, lines 89–92, line 95).

5. Page 5, lines 81-82. The Authors should report here and in the figure 2 legend how “low” and “high” annualized relapse rate were defined (not only in the methods section).

Author response and action taken: Also excellent feedback given the aforementioned likelihood that the article will be typeset with the figures and tables at the end of the paper rather than in-line with their supporting text. This suggestion has been incorporated (page 5, lines 102–103).

6. Page 5, lines 83-95. The Authors performed a Pearson correlation analysis which requires normal distribution and continuous nature of the two variables. Therefore, I did not agree to clip at five the number of lesions. In this analysis, Authors should use the unclipped number of lesions. Anyway, as I wrote before, this is not the appropriate analysis for the ARR endpoint.

Author response and action taken: To be clear, the regression analysis was for the count of Gd-enhancing lesions, not ARR. Also, recall that there were two regression analyses for Gd-enhancing lesion count as follows: a univariate one that searched for associations with single protein concentrations, and

a multivariate analysis that attempted to predict Gd-enhancing lesion count from an ensemble of protein concentration values.

Upon further reflection on your feedback, we agree that the univariate analysis does not need to be clipped at five lesions; the main reason we did this in the univariate analysis was for consistency with the multivariate one (which we will discuss at greater length below), but the analyses are totally independent of one another and there is no need for that beyond a rather aesthetic sense of symmetry. Furthermore, there is no reason to expect that any protein concentration should be linearly related to lesion count, an assumption further confounded by the fact that this analysis is done in NPX units (which is proportional to the log of protein concentration). For this reason, we have switched the univariate regression analysis from a Pearson's correlation to a Spearman's correlation, thus removing the assumption of linearity between the independent and dependent variables implicit in the Pearson's correlation. Dropping this implicit linearity assumption is another good reason to lift the univariate lesion count cap.

Regarding the multivariate lesion count regression, we clipped the analysis at five for the following reasons:

1. The overwhelming majority (473/501~0.944) of our Gd lesion samples have five or fewer lesions. Therefore, that is the region where we can trust and actually care about the behavior of this model. Furthermore, extending any predictions much above five lesions would be suspect since the model(s) that made them have not been trained on enough data in that range.
2. This problem is exacerbated by our cross-validation technique, since each of the 1000 bootstrap models built on this data sees a randomly selected 2/3 of the data set in training (and is evaluated on the remaining third). This means that many of these models are never trained on any sample with more than five sessions.
3. Furthermore, very high lesion counts are subject to rater-to-rater variation (not all of our samples had Gd lesions counted by the same neuroradiologist) that has a larger effect than for more modest counts. A rater with a propensity to split concave or multimodal lesions (e.g. dumbbell shaped or ones with multiple blobs joined by narrow connection regions) into multiple lesions can easily elevate a count of 5–8 lesions to something more like 10–15 depending on lesion morphology, scan contrast/signal-to-noise ratio, or any manner of other effects. The reverse argument is true for raters with the opposite proclivity.
4. The point of doing a regression analysis at all is to look at the smooth transition from patients with no Gd lesions to ones with a high burden. Clipping at five does just that, especially in light of the fact that in the “DMT era” of MS care, patients with very high lesion counts (>5) are increasingly rare (as seen in the abundances in our data).
5. Continuing the above point, it is interesting to note that our regression model actually mistakes a significant number of zero-lesion samples for one- (or even two-) lesion samples. This is one of the primary failure modes of the linear model we used for Gd lesion regression, making it an even worse idea to trust it in the increasingly rare case of lesion counts ranging up to 20 or more.

As a result, we simply treated five or more lesions essentially as an “overflow bin,” which just means that it was uncountably many for our analysis. To be clear, we are also keeping the Pearson’s correlation analysis in evaluating the multivariate regression, since it compares the predicted lesion count with the labeled one, making the linearity assumption baked into the Pearson’s correlation totally warranted.

7. Page 18, line 394. R2 is not the Pearson Correlation coefficient. R2 is the square of the Pearson Correlation coefficient.

Author response and action taken: Well spotted—thank you. We have made the necessary edit, and the passage was revised to the following (page 19, lines 428–430):

“Both the regression and classification analyses were performed on an optimized subset of the entire panel chosen using GFS of protein features, with R^2 and AUROC, respectively, as the target metric and average over the ensemble of bootstrap splits.”

REVIEWERS' COMMENTS

Reviewer #1 (Remarks to the Author):

My previous concerns and comments have been addressed exhaustively. The authors have made comprehensive amendments that have significantly improved the quality and clarity of the article. Therefore, I endorse this article for publication in its current form

Reviewer #3 (Remarks to the Author):

Authors sufficiently clarified all my previous concerns. The manuscript has been improved with many important technical details and descriptions.

Some easily addressable concerns still remain.

Study design: Authors better explained how the study population has been selected. But they did not report yet the study design. Please add explicitly at the beginning of the results section and in the methods if this is a cross-sectional study or a case-control study. The Nature Comm. Reader is used the read the type of the study design at issue. In the "Material" section of the manuscript is not still clear if patients were selected: all with available biological samples? Or the authors searched for cases and controls for each of the three endpoints?

Authors reported from the very beginning that they analysed 630 samples (that's true) but the analyses on the endpoints at issues were conducted using 506 patients for Gd lesion), 161 patients for dichotomized ARR and 124 patients for Clinically Active/Inactive endpoint. Please be clear from the beginning.

Page 7-8, lines 159-161. "Comparison of the actual versus estimated Gd lesion count (left panel, Fig. 5) showed that the regression model slightly under-predicts Gd lesion count for samples with no lesions...". Authors used a linear regression (developed for continuous dependent variables) for a count endpoint (Gd lesion count) and this choice resulted in a clinically non acceptable result: this model can predict a negative lesions count... what does it mean? A patient is predicted to have -2 lesions? This is why Poisson model, negative binomial model and other framework for count data have been developed and properly used in Multiple Sclerosis. Again, my suggestion is to appropriately re-run analyses on Gd lesion counts. Otherwise, you should use only the dichotomized version presence/absence of any Gd lesion.

Table 3. Please specify in the caption what "Low ARR" and "High ARR" mean.

REVIEWER COMMENTS

Reviewer 1 (Remarks to the Author):

My previous concerns and comments have been addressed exhaustively. The authors have made comprehensive amendments that have significantly improved the quality and clarity of the article. Therefore, I endorse this article for publication in its current form.

Author response and action taken: Thank you for your review and endorsement.

Reviewer 3 (Remarks to the Author):

Authors sufficiently clarified all my previous concerns. The manuscript has been improved with many important technical details and descriptions.

Some easily addressable concerns still remain.

1. Study design: Authors better explained how the study population has been selected. But they did not report yet the study design. Please add explicitly at the beginning of the results section and in the methods if this is a cross-sectional study or a case-control study. The Nature Comm. Reader is used the read the type of the study design at issue. In the “Material” section of the manuscript is not still clear if patients were selected: all with available biological samples? Or the authors searched for cases and controls for each of the three endpoints?

Author response and action taken: We now explicitly state that this was a cross-sectional analysis at the beginning of the Results section (page 4, line 82). We also remind readers of this in the Discussion section (page 11, line 251), as well as in the Methods section in both the Materials subsection (page 13–14, lines 300–301 and page 14, line 309), and the Biostatistical Methods subsection (page 15, line 345).

2. Authors reported from the very beginning that they analysed 630 samples (that’s true) but the analyses on the endpoints at issues were conducted using 506 patients for Gd lesion), 161 patients for dichotomized ARR and 124 patients for Clinically Active/Inactive endpoint. Please be clear from the beginning.

Author response and action taken: A more detailed discussion of the sample breakdown across studies and endpoints is now included at the end of the Materials subsection of the Methods section (page 14, lines 315–317). We also added a sentence (page 14, lines 309–311) stating that cross-sectional samples were taken from the three cohorts to balance occupancy across the gadolinium (Gd) lesion and clinical relapse status endpoints, which reads:

“Samples were chosen to balance occupancy across the Gd lesion and CRS endpoints (ARR represented more of an opportunistic endpoint).”

3. .Page 7-8, lines 159-161. “Comparison of the actual versus estimated Gd lesion count (left panel, Fig. 5) showed that the regression model slightly under-predicts Gd lesion count for samples with no lesions...”. Authors used a linear regression (developed for continuous dependent variables) for a count endpoint (Gd lesion count) and this choice resulted in a clinically non acceptable result: this model can predict a negative lesions count... what does it mean? A patient is predicted to have -2 lesions? This is why Poisson model, negative binomial model and other framework for count data have been developed and properly used in Multiple Sclerosis. Again, my suggestion is to appropriately re-run analyses on Gd lesion counts. Otherwise, you should use only the dichotomized version presence/absence of any Gd lesion.

Author response and action taken: We re-ran the Gd lesion regression analysis to use Poisson regression instead of Ridge regression for both greedy forward selection and the multivariate analyses. This avoids any samples where a negative number of samples was predicted by the multivariate model. The text of the Multivariate results subsection, plots/captions for Figures 4 and 5, as well as Table 2, have been updated to reflect changes to the analysis.

4. Table 3. Please specify in the caption what “Low ARR” and “High ARR” mean.

Author response and action taken: We have now clarified these terms.